# Platelet Rich Plasma Injections for Knee Osteoarthritis Treatment: A Prospective Clinical Study

**DOI:** 10.3390/jcm11092640

**Published:** 2022-05-08

**Authors:** Lorenzo Moretti, Giuseppe Maccagnano, Michele Coviello, Giuseppe D. Cassano, Andrea Franchini, Andrea Laneve, Biagio Moretti

**Affiliations:** 1Orthopaedic & Trauma Unit, Department of Basic Medical Sciences, Neuroscience and Sense Organs, School of Medicine, University of Bari Aldo Moro, AOU Consorziale Policlinico, 70124 Bari, Italy; lorenzo.moretti@libero.it (L.M.); michelecoviello91@gmail.com (M.C.); dancassanox@gmail.com (G.D.C.); andreafranchini1988@gmail.com (A.F.); biagio.moretti@uniba.it (B.M.); 2Orthopaedics Unit, Department of Clinical and Experimental Medicine, Faculty of Medicine and Surgery, University of Foggia, Policlinico Riuniti di Foggia, 71122 Foggia, Italy; andreaa.laneve@gmail.com

**Keywords:** knee osteoarthritis (KOA), cartilage, plateled riched plasma (PRP), knee injection, biologic therapy

## Abstract

Background: The aim of this prospective study was to evaluate the efficacy and safety of Platelet Rich Plasma (PRP) injections in patients affected by knee osteoarthritis (KOA). An autologous blood product containing a high percentage of various growth factors (GFs), cytokines and modulating factors as PRP has shown promising results in achieving this goal. Methods: One hundred and fifty-three patients (72 males, mean age 59.06 ± 8.78, range 40–81 years old) from January 2018 to January 2020 received three consecutive PRP injections and completed the follow ups. Western Ontario and McMaster University Osteoarthritis index (WOMAC), Knee society score (KSS) and Visual Analogic Scale (VAS) were evaluated before PRP injection (T0), one month (T1), three months (T2) and six months (T3) after the treatment. All patients underwent baseline and at 6 months MRI and X-ray evaluation. Results: A statistically significant VAS, KSS and WOMAC reduction emerged in the comparison between evaluations (*p* < 0.05), MRI demonstrated non-statistically significant improvement in cartilage thickness for both tibial plate and femoral plate (*p* = 0.46 and *p* = 0.33 respectively), and no radiographic changes could be seen in any patients. Conclusions: PRP injection represents a valid conservative treatment to reduce pain, improve quality of life and functional scores even at midterm of 6 months follow-up.

## 1. Introduction

Cartilage structure modifications are responsible for several degenerative joint diseases, such as chondropathy and osteoarthritis. Osteoarthritis (OA) is one of the most common progressive and degenerative knee diseases, affecting the intra-articular, tibiofemoral, and patellofemoral cartilage together with the adjacent joints and structures [1,2]. Musculoskeletal pain and movement restriction are symptoms associated with OA, resulting in a reduction in daily performance [3,4]. According to Kurtz et al. [5] the percentage of knee OA is expected to increase over the next 10 years due to the growing rate of obesity and of the population’s average age. Additionally, a recent study demonstrated that thickness and volume of cartilages are significantly lower for 50-year-old patients, with a Body Mass Index (BMI) equal to and greater than 25 [6]. Several surgical and nonsurgical treatments have been suggested to treat the join pain [7,8]; the latest are especially recommended for young and middle-aged patients presenting earlier stages of OA [1]. Among the conservative treatments, the use of non-steroidal anti-inflammatory drugs (NSAIDs), the intra-articular injections with corticosteroid (CS) or hyaluronic acid (HA) and saline have been used to manage mild KOA for several years [4]. Various meta-analyses investigated the effectiveness of the Platelet-rich plasma (PRP) by comparing it with other procedures, the results highlighted a better pain relief and functional improvement observed at different times after injection. In particular, the PRP is an autologous blood product containing a high percentage of various growth factors (GFs), such as fibroblast growth factor, epidermal growth factor, vascular endothelial growth factor, transforming growth factor-β and platelet-derived growth factor [4]. A recent study suggested that these GFs and cytokines, released by platelets after being damaged by an injury or pathology, might be involved in modulating the inflammatory processes contributing to the tissue structures preservation or regeneration [9]. Different APCs (Autologous platelet concentrates) were used in regenerative and reparative medicine, such as plasma rich in growth factors (PRGF), advanced-platelet rich fibrin (A-PRF) and injectable-platelet rich fibrin (i-PRF). They demonstrated excellent results like PRP in different fields such as maxillofacial surgery and orthopaedic. [10,11,12]

Moreover, the effect of PRP injections on MRI changes remains unclear [13]. The current study aims to assess the clinical effects of PRP injections in patients affected by KOA of grades from 1 to 3 (Kellgren-Lawrence (K-L) radiographic classification scale) at 6 months follow-up, with VAS reduction as a definite primary endpoint.

## 2. Materials and Methods

We designed a prospective study which was approved by the local Ethics Committee (delib. 0104) and registered at ClinicalTrials.gov (accessed on 21 April 2021) (NCT04852380). The subjects enrolled were informed that data from the research would be submitted for publication and signed an informed consent form.

Two hundred and ten patients referred to the Orthopaedic and Trauma Unit of the local University Hospital between January 2018 and January 2020 with knee osteoarthritis (KOA) were prospectively recruited. Twelve of them refused the procedures, twenty-one do not respect inclusion or exclusion criteria and fourteen were excluded because they were affected by SARS-COV2. Ten patients were lost at the follow up. Finally, 153 patients were enrolled in the study (Figure 1).

Inclusion criteria were: (1) Age between 40 and 81; (2) Body mass index (BMI) between 20 and 29.9; (3) Chronic history (for at least 4 months) of knee joint pain; (4) Radiographically documented knee osteoarthritis of grades 1 to 3 (Kellgren-Lawrence K-L radiographic classification scale).

Exclusion criteria were: (1) Radiographically severe documented knee osteoarthritis of grade 4 (K-L radiographic classification scale); (2) Previous femur and tibia fractures; (3) Knee previous surgical treatment (e.g., arthroscopy); (4) Hyaluronic acid infiltration within the previous six months; (5) Hemoglobin levels < 10 g/dL; (6) History of oncohematological disease, infections, or immunodepression.

All the patients were subjected to a clinical evaluation before starting the procedure at each visit, and routine blood tests were carried out before injection, including complete blood count and screening for transmittable diseases (e.g., HIV, HBV, HCV).

All tests were performed in the same place, and the same researchers (two orthopaedic surgeons with more than ten years of experience in knee surgery) tested all patients. The evaluation times were: T0 (recruitment), T1 (one month after the last injection), T2 (three month after the last injection), T3 (six month after the last injection). The Western Ontario and McMaster Universities Osteoarthritis Index (WOMAC), the Knee Score Society (KSS) score and the Visual Analogue Scale (VAS) were evaluated and recorded for each patient at each follow up. Their MRI and X-ray images at baseline and at 6 months were analyzed and included in this study.

As a primary endpoint, pain was quantified using the VAS scale with scores ranging from 0 (no pain) to 10 (worst imaginable pain). 

The functional recovery as a secondary endpoint was monitored using the WOMAC index and KSS. 

The PRP concentrate required for injection was prepared at the Immunohematology Department of Bari University Hospital by apheresis of venous blood from each patient. All the patients were advised to fast for 10 h before the blood collection in an effort to avoid any effects of food intake on PRP concentrate, meanwhile water intake was not restricted. 

Venous blood drawn from each patient was centrifuged at room temperature with Arthrex Angel System (Arthrex^®^, Naples, FL, USA) in order to separate the blood, the plasma, the buffy coat and residual red blood cells (RBCs). The total amount of the PRP concentrate obtained was divided in three part (about 5 mL each), and then stored at −40 °C at Blood Bank of Bari University Hospital.

The 5 mL 5% concentrate PRP was injected every week three times, starting from recruitment. The patient was in a supine position with a 90 degree knee flexion. All the procedures were carried out in an aseptic condition with a 21-gauge needle inserted into the antero-lateral soft spot area of the knee. The patient was observed for 30 min under medical care after the procedures and then they were discharged if no complications appear. A post-treatment therapy was prescribed to each patient consisting of an antibiotic cycle, functional rest for 24–48 h, paracetamol (in case of postprocedural pain) and local cryotherapy. No adverse event was observed in the treated data.

The WOMAC index, ranging between 0 and 96, consists of five items for pain (score range 0–20), two for stiffness (score range 0–8), seventeen for functional limitations (score range 0–68) and WOMAC total index, with lower WOMAC value indicating benefits after treatment. The KSS is divided in two parts: Knee Score and Function Score. The first includes measure of pain (maximum 50 points), range of motion (maximum 25 points), and anteroposterior and mediolateral stability (maximum 25 points), and flexion contracture, extension range, and alignment were evaluated. The second part consists of walking and stair climbing skills (maximum 50 points each); at the same time walking aids were considered. Scores of 80 to 100 were considered as excellent, 70 to 79 good, 60 to 69 fair, and less than 60 poor [14].

Magnetic resonance imaging data were acquired on Siemens MAGNETOM^®^ Essenza (Milan, Italy) 1.5 T, extremity coil, and used for image analysis: sagittal T1 spin echo. Patients remained in a supine position with a fully extended knee and the foot perpendicular to the MRI table. Femoro-tibial cartilage was divided into four plates by anatomic location: medial tibial, lateral tibial, medial femoral and lateral femoral. Three measurement points were taken for each plate: anterior, median and posterior. The mean cartilage thickness and standard deviation (SD) were calculated for tibial and femoral plates at baseline and at 24 weeks. 

A prospective clinical study was conducted. The data were collected and analyzed using SPSS (v 23; IBM^®^ Inc., Armonk, NY, USA). Descriptive statistics were calculated for the overall sample and for follow-up and pathology. Categorical variables were presented as numbers or percentages. Continuous variables not normally distributed were presented as median and range normally distributed variables were presented as mean and standard deviation. The Shapiro-Wilk test was used to test for normality of the data.

Mann-Whitney U tests or Kruskal-Wallis tests for group comparisons were conducted for follow-ups and pathologies, since the variables were not normally distributed.

A *p*-value of <0.05 was considered statistically significant.

## 3. Results

### 3.1. Patient Characteristics

One hundred and fifty-three patients (72 males) affected by knee osteoarthritis disorder were recruited in the current study. The mean age of the OA patient was 59.06 ± 8.78 SD with the patients’ age range between 40 and 81 years. The mean BMI of all treated patients was 25.4 ± 3.9 SD. 

Demographics of patients are shown in Table 1.

### 3.2. WOMAC 

Statistically significant differences (*p* ≤ 0.05) occurred in the 4-time WOMAC for the functional limitations, pain and total WOMAC index. The WOMAC functional limitations value demonstrated statistically significant reduction between T0 and T1 (21.61 ± 12.86 SD vs. 15.78 ± 9.67 SD, *p* ≤ 0.001), between T1 and T2 (15.78 ± 9.67 SD vs. 12.41 ± 7.36 SD, *p* ≤ 0.001, between T2 and T3 (12.41 ± 7.36 SD vs. 7.43 ± 5.28 SD, *p* ≤ 0.001) and between T0 and T3 (21.61 ± 12.86 SD vs. 7.43 ± 5.28 SD, *p* ≤ 0.001). 

The WOMAC pain value showed statistically significant reduction between T0 and T1 (6.53 ± 3.65 SD vs. 4.83 ± 3.2 SD, *p* ≤ 0.001), between T1 and T2 (6.53 ± 3.65 SD vs. 4.09 ± 2.63 SD, *p* ≤ 0.05), between T2 and T3 (4.09 ± 2.63 SD vs. 2.31 ± 2.17 SD, *p* ≤ 0.001) and between T0 and T3 (6.53 ± 3.65 SD vs. 2.31 ± 2.17 SD, *p* ≤ 0.001). 

The WOMAC total value showed statistically significant reduction between T0 and T1 (30.56 ± 17.55 SD vs. 22.44 ± 13.38 SD, *p* ≤ 0.001), between T1 and T2 (22.44 ± 13.38 SD vs. 18.10 ± 10.30 SD, *p* ≤ 0.001), between T2 and T3 (18.10 ± 10.30 SD vs. 10.69 ± 7.53 SD, *p* ≤ 0.001) and between T0 and T3 (30.56 ± 17.55 SD vs. 10.69 ± 7.53 SD, *p* ≤ 0.001).

The WOMAC stiffness value showed statistically significant reduction between T0 and T1 (2.42 ± 2.12 SD vs. 1.82 ± 1.64 SD, *p* ≤ 0.05), between T2 and T3 (1.60 ± 1.37 SD vs. 0.93 ± 1.05 SD, *p* ≤ 0.001) and between T0 and T3 (2.42 ± 2.12 SD vs. 0.93 ± 1.05 SD, *p* ≤ 0.001). No statistically significant difference was shown between T1 and T2 (1.82 ± 1.64 SD vs. 1.60 ± 1.37 SD, *p* = 0.33).

Results are illustrated in Figure 1.

### 3.3. KSS

Statistically significant differences (*p* ≤ 0.05) occurred in the 4-time KSS for the knee score, and functional score, excepting for the value comparing T1 and T2 (*p* > 0.05) for functional score only. 

The Knee Score, part of KSS, showed a significant increase over time between T0 and T1(82.61 ± 16.24 SD vs. 87.50 ± 14.14 SD, *p* ≤ 0.001), between T1-T2 (87.50 ± 14.14 SD vs. 89.78 ± 12.41 SD, *p* ≤ 0.05), between T2-T3 (89.78 ± 12.41 SD vs. 91.01 ± 12.03 SD, *p* ≤ 0.05) and between T0-T3 (82.61 ± 16.24 SD vs. 91.01 ± 12.03 SD, *p* ≤ 0.001).

The functional KSS score showed a statistically significant differences between T0 and T1(82.97 ± 17.59 SD vs. 88.75 ± 14.91 SD, *p* ≤ 0.001), between T2-T3 (90.89 ± 13.15 SD vs. 93.72 ± 11.60 SD, *p* ≤ 0.05) and between T0-T3 (82.97 ± 17.59 SD vs. 93.72 ± 11.60 SD, *p* ≤ 0.001). No statistically significant difference was shown between T1 and T2 (88.75 ± 14.91 SD vs. 90.89 ± 13.15 SD, *p* = 0.10).

Results are illustrated in Figure 2.

### 3.4. VAS

The VAS score improved statistically significantly between T0 and T1 (4.81 ± 2.11 SD vs. 3.25 ± 1.85 SD, *p* ≤ 0.001), between T2 and T3 (2.77 ± 1.51 SD vs. 1.79 ± 1.51 SD, *p* ≤ 0.001), and between T0 and T3 (4.81 ± 2.11 SD vs. 1.79 ± 1.51 SD, *p* ≤ 0.001).

No statistically significant difference was shown comparing T1 and T2 (3.25 ± 1.85 SD vs. 2.77 ± 1.51 SD, *p* > 0.05).

Results are illustrated in Figure 3.

### 3.5. MRI

The tibial and femoral plates thickness improved non-statistically significantly between T0 and T3 (13.04 ± 2.64 SD vs. 13.23 ± 1.87 SD, *p* = 0.46 and 14.16 ± 2.56 SD vs. 14.40 ± 1.90 SD, *p* = 0.33, respectively) (Figure 4a,b). 

No X-ray changes have been demonstrated using Kellgren-Lawrence (K-L) radiographic classification scale (*p* > 0.05).

## 4. Discussion

The results of this study highlight that PRP infiltrations represent a useful conservative treatment to reduce pain, improve quality of life and functional scores at the midterm of 6 months follow-up in patients with knee osteoarthritis. The PRP injection benefits in joint disease are demonstrated by several authors [4,15,16]. Positive effects were shown on patients with decreasing pain as indicated by WOMAC pain index, KSS index and VAS. The WOMAC and KSS also indicated that the physical function was improved, even if there was no statistical evidence comparing the KSS score at 1 month and 3 months of the follow-up. A statistically significant subjective and objective improvement of the three scales was demonstrated. The effects of PRP injection on pain reduction have been previously observed in other studies and several authors [9,17,18] have reported the analgesic properties of platelets. More recently, a meta-analysis [2] indicated that PRP reduces pain by influencing the expression of mediators (e.g., prostaglandin E2, substance P, dopamine, 5-hydroxy-tryptamine) and that the GFs, contained in the PRP concentrate, promote the synthesis of cartilage matrix, stimulating the growth of chondrocytes and the inhibition of the local inflammatory response [19]. 

Tang et al. in their meta-analysis found the differences between the treatment of osteoarthritis with PRP versus Hyaluronic acid [20]. Compared with acid hyaluronic, PRP has greater benefits in the conservative treatment of knee osteoarthritis, such as less long-term discomfort and better knee joint function. PRP poses no additional risks and can be used as a conservative treatment for osteoarthritis in the knee.

Patients with knee osteoarthrosis who received PRP intra-articular injections had the best overall success when compared to steroids, hyaluronic acid, and placebo at 3, 6, and 12-month follow-ups [21].

Cavazos et al. [22] suggest that while both single and multiple PRP injections improved pain and there was no difference between the two, triple PRP injections were more effective than single injections in enhancing joint functionality in individuals with knee OA.

Moreover, different APCs (Autologous platelet concentrates) were used in regenerative and reparative medicine. For instance, advanced platelet-rich fibrin+ (A-PRF+) and leucocyte platelet-rich fibrin protocol (L-PRF) demonstrated to control bleeding and to promote injured tissues healing [23]. Platelet concentrate without the use of anticoagulants (i-PRF) showed a high influence on osteoblast behavior by influencing human osteoblast migration, proliferation and differentiation [24].

All the scores referring to stiffness and physical functional showed an improvement overtime, agreeing with a previous study [25,26] which pointed out a decrease in the WOMAC index and an increase in the KSS total score, suggesting a positive influence of the treatment. However, it is worth noting the lack of statistical differences in the Knee Score part of KSS that refers to flexion contracture, the extension lag, the alignment, anteroposterior and mediolateral stability in contrast with the pain and range of motion. This aspect has never been pointed out by other authors, which referred only to the total KSS score, suggesting that further study is required to highlight the role of the pain score on the total KSS score. The literature remains unclear about the correlation between MRI and PRP injections [13]. Most of the studies do not show an improvement in both RX and MRI [27], in line with our results. Moreover, a lack of standardization of cartilage thickness measurement, which is different for each study, is highlighted. A standardized method is needed to make the studies homogeneous and consequently to analyze the results with higher level studies, such as reviews or metanalysis.

This study has some limitations. First, a control group was not included in this study. Second, there is no comparison between single and multiple injections. Additionally, the follow-up period was short and long-term effectiveness was therefore not assessed. On the other hand, a strong point of this work is the selection of treated patients, respecting stringent inclusion and exclusion criteria and the sample size.

## 5. Conclusions

The results confirm the efficacy of the PRP injections on the KOA, also suggesting that decreasing pain was obtained already after one month after injection with best results observed after 6 months. They suggests in further studies that data could be collected and recorded at only three different times: at recruitment, one, and six months after administration, leading to a time reduced follow-up protocol.

Additionally, further studies are required to assess the long-term effects of this technique, testing the PRP injection on a large number of patients.

## Data Availability

The data presented in this study are available on request from the corresponding author. The data are not publicly available due to privacy.

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
