# Peer review of "Platelet Rich Plasma Injections for Knee Osteoarthritis Treatment: A Prospective Clinical Study"

_jcm, 2022, doi:10.3390/jcm11092640_

Round 1
Reviewer 1 Report
The manuscript entitled “Platelet Rich Plasma Injections For Knee Osteoarthritis treatment. A prospective clinical study” submitted to JCM aims to evaluate the efficacy and safety of PRP injections in patients affected by KOA.
The manuscript appears interesting however, many manuscripts on KOA topical medication are documented in the literature.
I have some suggestions to improve deeply the quality of the manuscript, enriching the text with further notions.
English form: I suggest to perform a check of English text form
Abstract: Please, structure in a better way this section. In particular, reduce background section.
Introduction: Before the specific part on PRP, I suggest introducing a short part on the various APCs (Autologous platelet concentrates) used in regenerative and reparative medicine, including references. In particular focusing on the healing properties [PMID: 33536010 - PMID: 33433526 - PMID: 30231809].
Aim: authors stated in the right way this part however, I suggest to specify the primary outcome variable.
Methods: I suggest to add a graphic that resume this part “Two hundred and ten patients referring to the Orthopaedic and Trauma Unit of the 66 local University Hospital between January 2018 and June 2020 with knee osteoarthritis 67 (KOA) were prospectively recruited. Twelve of them refused the procedures, twenty-one 68 do not respect inclusion or exclusion cryteria and fourteen were excluded because affected 69 by SARS-COV2. Ten patients were lost at the follow up. Finally, 153 patients were enrolled 70 in the study”.
I recommend moving the PRP preparation part before the analysed variables.
Discussion: introduce a brief part on APC properties in medicine reparative processes and heamostasis [PMID: 34775872 - PMID: 32613433 - PMID: 28351189]
After a careful review, this manuscript must to be re-evaluated
Thank you for this opportunity.
Reviewer 2 Report
'Platelet Rich Plasma Injections For Knee Osteoarthritis treatment. A prospective clinical study' is a well written paper on an interesting topic which needs a major revision.
The authors have registered the study at ClinicaTrials.gov (NCT04852380). Registering the study is a very correct research practice which deserves scientific recognition, however there is a misalignment in what is reported in the paper versus what is reported on ClinicaTrials.gov (NCT04852380).
The authors write that:"...Two hundred and ten patients referring to the Orthopaedic and Trauma Unit of the 66 local University Hospital between January 2018 and June 2020 with knee osteoarthritis 67 (KOA) were prospectively recruited....".
However, on the website ClinicaTrials.gov (NCT04852380) they report the following:
- Actual Study Start Date : January 1, 2018
- Actual Primary Completion Date : January 1, 2020
The Actual Primary Completion Date is PRIOR to the end of the recruitment, which is obviously impossible.
In fact the US FDA and ClinicalTrials.gov definition is:
Primary completion date
The date on which the last participant in a clinical study was examined or received an intervention to collect final data for the primary outcome measure. Whether the clinical study ended according to the protocol or was terminated does not affect this date. For clinical studies with more than one primary outcome measure with different completion dates, this term refers to the date on which data collection is completed for all the primary outcome measures. The "estimated" primary completion date is the date that the researchers think will be the primary completion date for the study.
I think the authors should reconcile the dates on the paper, on the Ethical Committee reports and on ClinicalTrials.gov carefully before publishing.
Round 2
Reviewer 1 Report
The authors improved the quality of the manuscript by introducing more explanatory graphics. Although the topic of the manuscript is not new, with numerous studies present in the literature.
Reviewer 2 Report
The authors updated the paper based on the observations. The observed conflict between the Actual Primary Completion Date : January 1, 2020 reported on ClinicalTrials.gov versus June 2020 reported on the original submission has been corrected by the authors as a simple typo in the paper itself.